# Nasal Mycology of Chronic Rhinosinusitis Revealed by Nanopore Sequencing

**DOI:** 10.3390/diagnostics12112735

**Published:** 2022-11-09

**Authors:** Rong-San Jiang, Chien-Hung Shih, Yu-Han Jiang, Han-Hsueh Hsieh, Yi-Fang Chiang, Han-Ni Chuang, Tzu-Hung Hsiao

**Affiliations:** 1Department of Medical Research, Taichung Veterans General Hospital, Taichung 40705, Taiwan; 2Precision Medicine Center, Taichung Veterans General Hospital, Taichung 40705, Taiwan; 3Department of Otolaryngology, Taichung Veterans General Hospital, Taichung 40705, Taiwan; 4School of Medicine, Chung Shan Medical University, Taichung 40201, Taiwan; 5Rong Hsing Research Centre for Translational Medicine, National Chung Hsing University, Taichung 40227, Taiwan; 6Department of Critical Care Medicine, Taichung Veterans General Hospital, Taichung 40705, Taiwan; 7School of Medicine, College of Medicine, National Yang Ming Chiao Tung University, Taipei 112304, Taiwan; 8Department of Public Health, College of Medicine, Fu Jen Catholic University, New Taipei City 242062, Taiwan; 9Institute of Genomics and Bioinformatics, National Chung Hsing University, Taichung 40227, Taiwan

**Keywords:** chronic rhinosinusitis, fungal culture, fungus, nanopore sequencing

## Abstract

Background: Nanopore sequencing (NS) is a third-generation sequencing technology capable of generating reads of long sequences. In this study, we used NS to investigate nasal mycology in patients with chronic rhinosinusitis (CRS). Methods: Nasal cavities of 13 CRS patients were individually irrigated with 20 mL of distilled water. The irrigant was forcefully blown by the patient into a basin. The collected fluid was placed into a centrifuge tube and processed using the method of Ponikau et al. The collected specimens were used for traditional fungal culture and sequenced for total DNA using NS. Results: Traditional fungal culture successfully grew fungi in the specimens of 11 (84.6%) patients. *Aspergillus* sp. and *Penicillium* sp. were found in four (30.8%) patients, *Cladosporium* sp. in three (23.1%) patients, and Candida albicans, *Mucor* sp. and *Chaetomium* sp. in one patient. NS revealed fungi abundance ranged from 81 to 2226, with the Shannon species diversity ranging from 1.094 to 1.683 at the genus level. *Malassezia* sp. was sequenced in 13 patients, *Aspergillus* sp. in 12 (92.3%) patients, Candida albicans in 11 (84.6%) patients, and *Penicillium* sp. in 10 (76.9%) patients. Conclusion: Our results showed that NS was sensitive and fast in detecting nasal fungi in CRS patients.

## 1. Introduction

Chronic rhinosinusitis (CRS) is an inflammatory disorder of the paranasal sinuses and linings of the nasal passages, with the persistence of characteristic signs and symptoms lasting longer than 12 weeks [1]. The etiology of CRS is multifactorial, including infection, anatomic anomaly, allergy and genetic [2,3,4].

Over the last 20 years, it has also been suggested that fungi causes CRS by dysregulating the immune response, inducing the breakdown of the epithelial membrane, and exacerbating local inflammation of sinonasal mucosa [5,6,7]. The ubiquitous presence of fungi in CRS patients has been demonstrated in several studies [8,9,10]. These studies employed modified traditional culture techniques (e.g., method of Ponikau et al. [8]) to detect fungi. Although Ponikau et al.’s method [8] is more sensitive than traditional culture methods, culture-based methods are still limited by selective pressures of the nutrient media. Moreover, only 1% to 10% of known microorganisms are presumed to be culturable in the laboratory [11].

Culture-independent techniques, like PCR assays, potentially detect all organisms in CRS patients [12]. However, PCR uses specific primers or probes, targeting species to detect a limited number of pathogens. Recently, metagenomic sequencing detected all DNA present in a sample, enabling analysis of the entire microbiome, as well as the human host genome [13]. Previous studies have used next-generation sequencing (NGS) techniques to detect the microbiota in CRS patients [14,15].

Nanopore sequencing (NS) (Oxford Nanopore Technologies, Oxford, UK) is a third generation (i.e., single-molecule) sequencing technology thatgenerates long sequence read-lengths [16].16 It is characterized by culture-free, fast, single-molecule sequencing and sequencing in real-time [17]. The long sequence read-lengths requires no PCR amplification of the template. It minimizes bias during the library construction, as in the case of PCR [17]. NS was reported to identify, in real-time, species of fungi in dogs [18]. Here, we aimed to use NS to characterize the fungal microbiome of the nasal cavity in CRS patients.

## 2. Materials and Methods

### 2.1. Patients

We included CRS patients who had failed medical treatment and underwent bilateral primary functional endoscopic sinus surgery (FESS) between September 2018 and September 2019. CRS was diagnosed according to the EPOS criteria based on history, and findings in endoscopy and CT imaging [19]. Sinonasal inflammation was defined as having characteristic symptoms that persisted longer than 12 weeks, and both endoscopic examinations and CT scans showed supporting evidence. We excluded patients aged < 20 years old, those that had a history of immunodeficiency, and those who had taken antibiotics within a week before FESS. Those who were diagnosed pathologically as having sinonasal tumors or fungal sinusitis were also excluded. All eligible patients underwent nasal irrigation to collect nasal secretion for the modified traditional fungal culture (Ponikau et al.’s method [8]) and NS on the day before surgery. This study was approved by the Institutional Review Board (I) of Taichung Veterans General Hospital (protocol code CF17328B). Written consent was obtained from each patient.

### 2.2. Fungal Culture Using Ponikau et al’s Method

Patients were instructed to inspire deeply and to hold that position. One nasal cavity was irrigated by a syringe containing 20 mL of sterile distilled water. The irrigant was forcefully exhaled by the patient into a sterile pan. The irrigant in the pan was poured into a centrifuge tube and transferred to the microbiology laboratory. Under a laminar flow hood, an equal volume of diluted dithiothreitol (1.055 mg/mL) was added to the centrifuge tube. The tube was vortexed for 30 s and placed at room temperature for 15 m to allow the dithiothreitol to break down the disulfide bonds in the mucus. Then, the tube was centrifuged at 3000× *g* for 10 m and the supernatant was discarded. The sediment in the tube was vortexed for 30 s and the sediment was divided into two parts. One part was inoculated onto a Sabouraud dextrose agar plate and a Sabouraud dextrose agar plate containing chloramphenicol and cycloheximide. The agar plates were incubated at 30 °C and examined for 30 days on a daily basis. All isolates were identified. The other part was transferred for NS.

### 2.3. DNA Extraction from the Nasal Irrigant

Using sterile tips, the sediment of the nasal irrigant was transferred into a tube containing 100–200 μL of sterile water, which was centrifuged for 5 s at high speed to pellet cells. The cell suspension and lytic enzyme solution were mixed by inverting 25 times and incubated for 30 m at 37 °C. The remaining procedure of the extraction was in accordance with instructions of the Puregene yeast/bacteria kit B (Qiagen cat. 1042607). The extracted DNA was stored at −80 °C.

### 2.4. PCR-Free Library Preparation and NS

In the laboratory, 0.1–0.2 μg of the extracted DNA was packaged into the library for the NS system. DNA libraries were prepared according to the manufacturer’s instructions, using the ligation sequencing kit (SQK-LSK109) and the native barcoding kit (EXP-NBD104), including the optimization DNA sequence of the KAPA Hyper Prep Kit. The MinION (Oxford Nanopore Technologies, Oxford, UK) flow cell preparation and sample loading were in accordance with the 75 μL DNA library of the SQK-LSK109 protocol for the sequencing. The sequencing mixture was added into the R9.4.1 or R10 flow cell for 48–72 h.

### 2.5. Bioinformatic Analysis

NS from Oxford Nanopore Technologies included a real-time analysis with the EP2MI platform ‘what’s In My Pot’ (WIMP). For further in-depth analyses, the fast5 or fastq files with the sequencing reads were basecalled. Barcodes and adapters were removed using the porechop (https://github.com/rrwick/Porechop (accessed on 19 October 2018)). Taxonomy was assigned with the cloud-based analysis WIMP software application from the EPI2ME platform (Oxford Nanopore Technologies Ltd., UK) based on the Centrifuge (https://ccb.jhu.edu/software/centrifuge/manual.shtml (accessed on 5 June 2018)). The R9.4.1 flow cells (Oxford Nanopore Technologies Ltd., UK) were loaded with 75 μL of DNA library. The 18s rRNA and ITS gene sequence libraries were prepared with the kit according to the standard procedures, described by Oxford Nanopore Technologies Ltd., UK. The complete 18 s RNA gene was amplified using LongAmp^®^Taq 2X Master Mix (New England Biolabs, Ipswich, MA, USA) with the barcoded nanopore sequence primers (27F 5′-AGA GTT TGA TCM TGG CTC AG-3′ and 149R 5′-CGG TTA CCT TGT TAC GAC TT-3′). DNA amplification was performed on a T100 Thermal Cycler (Biorad, Lunteren, The Netherlands) using the following procedure: 1 m denaturation at 95 °C, 25 cycles (95 °C—20 s, 55 °C—30 s, 68 °C—2 m) and a final extension step of 5 m at 65 °C. The 16 S rRNA gene amplicons were quantified using Quant-IT™ PicoGreen™ (Thermo Fisher Scientific, Breda, The Netherlands). Equal amounts of amplicons per sample were pooled and the library was further processed following manufacturer’s instructions. Next, the library was incubated with Library Loading Beads (Oxford Nanopore Technologies, Oxford, UK) and the mixture was added to the MinIon/GridIon flow cell (version R9.2 or R.9.4, Oxford Nanopore Technologies, Oxford, UK) (Figure 1).

## 3. Results

### 3.1. Clinical Characteristics of Patients

In this study, we collected 13 CRS patients. Their demographic data are shown in Table 1. There were nine males and four females, with a mean age of 48.8 years old (range: 21 to 80 years). There was one smoker, but none suffered from asthma. Three patients had atopic dermatitis, and three used nasal steroids before enrollment. The severity of rhinosinusitis on the irrigated side of the nasal cavity was evaluated using the Lund-Kennedy endoscopic scoring system [20] and the Lund-Mackay CT scoring system [21]. Their endoscopic scores ranged from 2 to 5 at a mean of 3.4, and the CT scores ranged from 5 to 11 at a mean of 7.7. Among these CRS patients, six had nasal polyps, and seven had no nasal polyp. The surgical specimens of FESS showed that eight patients were eosinophilic CRS(tissue eosinophils > 10 cells per high power field) and five were non-eosinophilic CRS [22].

### 3.2. Isolation of Fungi Using Ponikau et al’s Method

Using Ponikau et al.’s method [8], 11 of 13 (84.6%) specimens grew fungi of one to three species. The most common fungi were *Aspergillus* sp. and *Penicillium* sp. cultured from four (30.8%) patients. *Cladosporium* sp. was isolated from three (23.1%) patients. *Candida albicans*, *Mucor* sp., *Chaetomium* sp., and an unidentified mold grew in one (7.7%) patient (Table 1).

### 3.3. Identification of Fungi by Nanopore Sequencing

The nanopore output total sequence reads were 91,476 to 1,598,608, and human read counts accounted for 28.35% to 96.68% (Table 2). However, the nanopore output sequence had fungal read counts that only accounted for 0.02 to 0.47% (Table 2). In this study, we only focused and analyzed fungal DNA. Nanopore sequence reads of fungi were from 54 to 2219 (Table 2). In 13 CRS patients, 36 to 2226 operational taxonomic units (OUTs) were identified, and the Shannon species diversity was from 1.094 to 1.683 at the genus level (Table 3). Ten most abundant fungal genera accounted for 56.15% to 89.13% of the nasal microbiota (Table 3). At the genus level, NS-identified major fungi were *Malassezia*, *Verticillium*, *Phycomyces*, and *Lobosporangium* et al (Figure 2).

Using NS, *Malassezia* sp. was identified in 13 (100%) of the specimens, *Aspergillus* sp. in 12 (92.3%), *Candida albicans* in 11, and *Penicillium* sp., *Chaetomium* sp. in 10 (Table 4). Thus, NS was more sensitive in detecting fungal species. However, *Cladosporium* sp. and *Mucor* sp. were not detected by NS (Table 4). We detected one to five pathogenic fungi in CRS patients using NS, such as *Malassezia*, *Aspergillus*, *Penicillium*, *Candida albicans* and *Chaetomium* (Table 5). The relative abundance of *Malassezia* sp. was from 12.34 to 52.08% and *Aspergillus* sp. from 0.8 to 4.26% (Table 3).

## 4. Discussion

Fungi have been reported to be ubiquitous in the sinuses of healthy subjects and CRS patients [8,9,10]. However, fungi are difficult to grow on the culture plate. Culture-negativity does not mean absence of fungi in the clinical samples [23]. We have applied third generation sequencing to evaluate fungi communities in 13 CRS patients with fungal rhinosinusitis. With NS, we found in all 13 samples of nasal irrigant the presence of three major pathogen species including *Malassezia* sp., *Aspergillus* sp. and *Candida albicans.* In contrast, the traditional culture method yielded only one or two species, and the growth process took several weeks. Therefore, we concluded that NS was more sensitive and faster in detecting fungi compared to the traditional culture method [24].

The sequencing techniques (NGS and third-generation sequence) are culture-independent. They have contributed to the knowledge of the comprehensive microbial communities in the human body. Modern technological advances in NGS allow the cost-effective assessment of microbial communities (microbiota) from environmental samples. Specifically, targeted sequencing of taxonomically informative regions of the genome (such as the 16S rDNA gene and internal transcribed spacer regions) reliably identifies most bacteria and fungi, down to the genus level [25].

Recently, metagenomic sequencing has been used to characterize the CRS mycobiome. One study on CRS patients, using NGS, reported nasal fungi as identified through sequencing the internal transcribed spacer region [26]. The most frequently detected fungi were *Aspergillus* sp., *Schizophyllum* sp., *Curvularia* sp., and *Malassezia* sp. Our study is the first to use NS to detect nasal fungi in CRS patients. We found the most frequently detected fungi were *Malassezia* sp., *Aspergillus* sp., *Candida albicans*, *Penicillium* sp. and *Chaetomium* sp. in nasal samples of 13 CRS patients. Our data showed that the NS was real-time and faster than the NGS in detecting microbiota of CRS. While NGS needs to collect a panel of specimens before sequencing, NS needs only one specimen to do the sequencing.

## 5. Conclusions

NS was found to be faster than the culture method in detecting fungi in nasal specimens of CRS patients, yielding more species of pathogenic fungi.

## Figures and Tables

**Figure 1 diagnostics-12-02735-f001:**
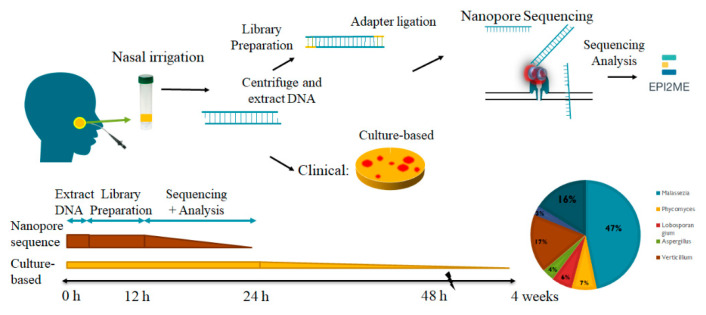
Schematic representation of the culture-based method and oxford nanopore sequencing process. The clinical culture-based system needs more than one month for fungal growth. The nanopore sequencing system only needs 24 h to identify fungi.

**Figure 2 diagnostics-12-02735-f002:**
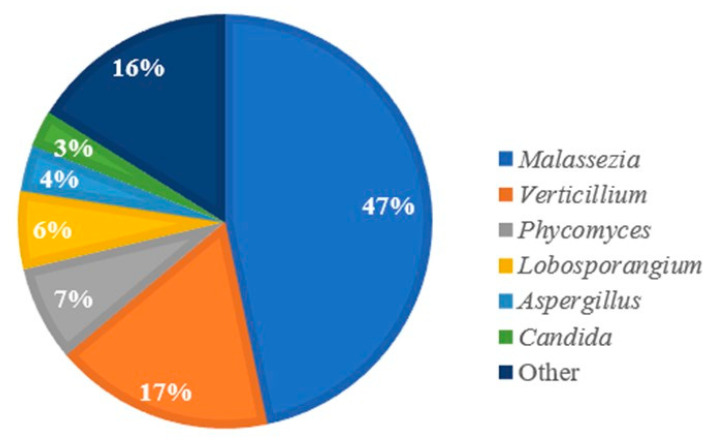
A Pie chart of the proportion abundance of fungi identified through nanopore sequencing at the genus level.

**Table 1 diagnostics-12-02735-t001:** Demographic data of chronic rhinosinusitis patients.

Patient	Sex	Age	Smoking History	Nasal Steroid	Atopic Dermatitis	Nasal Polyps	Eosinophilic CRS	Endoscopic Score	CT Score	Culture Result
1	F	44	N	N	N	Y	N	4	9	*Mucor* species
2	F	73	N	N	N	N	N	3	7	*Aspergillus fumigatus*
										*Aspergillus niger*
										*Cladosporium* species
3	M	39	N	N	Y	N	Y	3	7	*Penicillium* species
4	M	51	Y	N	N	Y	Y	4	11	*Aspergillus niger*
										*Penicillium* species
5	M	27	N	N	Y	N	N	3	5	*-*
6	M	61	N	N	Y	N	N	4	8	*Unidentified mold*
7	M	80	N	Y	N	Y	Y	5	11	*-*
8	M	53	N	Y	N	Y	Y	3	7	*Candida albicans*
9	M	45	N	Y	N	N	Y	4	6	*Aspergillus niger*
10	M	21	N	N	N	Y	Y	3	6	*Aspergillus flavus*
										*Cladosporium* species
11	F	54	N	N	N	N	N	2	8	*Cladosporium* species
12	M	45	N	N	N	N	Y	2	6	*Chaetomium* species
										*Penicillium* species
13	F	41	N	N	N	Y	Y	4	9	*Penicillium* species

CRS, chronic rhinosinusitis; CT, computed tomography; F, Female; M, Male; Y, yes; N, No.

**Table 2 diagnostics-12-02735-t002:** Summary nanopore output total sequence reads and percentage of species form the nasal irrigant in patients of chronic rhinosinusitis.

Patient	Total Reads	Reads Classified	Reads Unclassified	Human Read Counts	% of Human Read	Fungal Read Counts	% of Fungal Read	Bacterial Read Counts	% of Bacterial Read	Archaeal Reads	Viral Reads
1	91,476	86,209	5267	85,768	93.76%	105	0.11%	246	0.27%	1	2
2	260,652	247,711	12,941	246,976	94.75%	276	0.11%	257	0.10%	3	5
3	124,092	119,782	4310	119,206	96.06%	125	0.10%	378	0.30%	0	2
4	461,770	438,200	23,570	437,483	94.74%	309	0.07%	121	0.03%	4	2
5	473,888	454,052	19,836	449,829	94.92%	2219	0.47%	689	0.15%	4	191
6	587,767	553,744	34,023	548,601	93.34%	675	0.11%	3763	0.64%	9	12
7	673,783	655,678	18,105	651,395	96.68%	347	0.05%	2499	0.37%	4	34
8	944,779	919,864	24,915	827,692	87.61%	912	0.10%	64,239	6.80%	4	244
9	1,598,608	1,537,782	60,826	1,389,765	86.94%	1458	0.09%	101,373	6.34%	5	379
10	598,282	577,110	21,172	530,416	88.66%	634	0.11%	34,257	5.73%	2	156
11	339,267	328,838	10,429	226,793	66.85%	273	0.08%	69,760	20.56%	2	314
12	228,665	226,967	1698	64,832	28.35%	54	0.02%	128,244	56.08%	0	348
13	516,000	483,958	32,042	314,131	60.88%	470	0.09%	107,473	20.83%	5	783

**Table 3 diagnostics-12-02735-t003:** Fungi identified in the nasal irrigation of chronic rhinosinusitis based on nanopore sequencing.

Patient	Shannon Species Diversity	Number of Fungi Genus Identified (OTU)	Top 10 Genus									
			1	2	3	4	5	6	7	8	9	10
1	1.672	81	*Malassezia*	*Phycomyces*	*Lobosporangium*	*Aspergillus*	*Coccidioides*	*Marssonina*	*Sclerotinia*	*Capronia*	*Penicilliopsis*	*Paracoccidioides*
			35.80%	17.28%	8.64%	4.94%	3.70%	3.70%	3.70%	2.47%	2.47%	2.47%
2	1.297	242	*Malassezia*	*Phycomyces*	*Lobosporangium*	*Verticillium*	*Aspergillus*	*Bipolaris*	*Mitosporidium*	*Penicillium*	*Sclerotinia*	*Colletotrichum*
			43.39%	7.02%	4.96%	3.72%	3.31%	2.07%	2.07%	1.65%	1.65%	1.65%
3	1.445	92	*Malassezia*	*Phycomyces*	*ParaCoccidioides*	*Tuber*	*Marssonina*	*Verticillium*	*Lodderomyces*	*Laccaria*	*Melampsora*	*Lobosporangium*
			54.35%	7.61%	4.35%	3.26%	3.26%	3.26%	3.26%	3.26%	3.26%	3.26%
4	1.321	276	*Malassezia*	*Phycomyces*	*Colletotrichum*	*Lobosporangium*	*Aspergillus*	*Lodderomyces*	*Penicilliopsis*	*Coccidioides*	*Metarhizium*	*Trichoderma*
			36.23%	6.88%	4.71%	3.62%	3.26%	2.90%	1.81%	1.81%	1.81%	1.81%
5	1.516	2226	*Verticillium*	*Malassezia*	*Lobosporangium*	*Phycomyces*	*Pestalotiopsis*	*Penicillium*	*Beauveria*	*Aspergillus*	*Tetrapisispora*	*Metschnikowia*
			30.10%	13.88%	4.67%	3.86%	3.82%	3.28%	3.05%	2.52%	2.38%	2.29%
6	1.23	644	*Malassezia*	*Phycomyces*	*Lobosporangium*	*Aspergillus*	*Verticillium*	*ParaCoccidioides*	*Ascoidea*	*Trichoderma*	*Candida*	*Anthracocystis*
			47.20%	4.97%	4.81%	3.26%	2.80%	2.64%	2.02%	1.55%	1.55%	1.55%
7	1.283	312	*Malassezia*	*Phycomyces*	*Lobosporangium*	*Aspergillus*	*Blastomyces*	*Verticillium*	*Trichoderma*	*Lodderomyces*	*Capronia*	*Penicilliopsis*
			43.91%	7.05%	5.13%	3.53%	2.24%	2.24%	2.24%	1.92%	1.60%	1.60%
8	1.128	888	*Malassezia*	*Verticillium*	*Phycomyces*	*Lobosporangium*	*Paracoccidioides*	*Aspergillus*	*Candida*	*Penicillium*	*Sclerotinia*	*Colletotrichum*
			53.49%	6.42%	4.95%	3.60%	1.58%	1.46%	1.35%	1.24%	1.24%	1.24%
9	1.094	1439	*Malassezia*	*Phycomyces*	*Lobosporangium*	*Verticillium*	*Aspergillus*	*Paracoccidioides*	*Colletotrichum*	*Trichoderma*	*Candida*	*Anthracocystis*
			47.12%	4.59%	2.78%	2.43%	2.15%	1.88%	1.67%	1.53%	1.39%	1.39%
10	1.683	608	*Verticillium*	*Malassezia*	*Candida*	*Colletotrichum*	*Phycomyces*	*Isaria*	*Penicillium*	*Aspergillus*	*Lobosporangium*	*Metarhizium*
			18.75%	18.42%	9.38%	6.58%	5.92%	3.62%	2.80%	2.63%	2.63%	1.97%
11	1.441	259	*Malassezia*	*Candida*	*Colletotrichum*	*Phycomyces*	*Lobosporangium*	*Isaria*	*Anthracocystis*	*Marssonina*	*Setosphaeria*	*Metarhizium*
			36.29%	8.49%	8.11%	4.25%	3.86%	2.70%	2.70%	1.93%	1.54%	1.54%
12	1.358	55	*Malassezia*	*Phycomyces*	*Anthracocystis*	*Lobosporangium*	*Schizosaccharomyces*	*Arthrobotrys*	*Exophiala*	*Penicilliopsis*	*Nannizzia*	*Trichophyton*
			50.91%	7.27%	3.64%	3.64%	1.82%	1.82%	1.82%	1.82%	1.82%	1.82%
13	1.52	447	*Malassezia*	*Lobosporangium*	*Phycomyces*	*Candida*	*Aspergillus*	*Colletotrichum*	*Lodderomyces*	*Metarhizium*	*Histoplasma*	*Isaria*
			12.98%	8.95%	7.83%	7.38%	4.47%	4.47%	3.13%	2.46%	2.24%	2.24%

OUT, Operational taxonomic unit.

**Table 4 diagnostics-12-02735-t004:** Comparison of the genus level of fungi between Ponikau et al.’s method [8] and nanopore sequencing.

Genus	Ponikau et al.’s Method [8] (N, %)	Nanopore Sequencing (N, %)
*Malassezia* sp.	-	13 (100%)
*Aspergillus* sp.	4 (30.8%)	12 (92.3%)
*Penicillium* sp.	4 (30.8%)	10 (76.9%)
*Cladosporium* sp.	3 (23.1%)	-
*Candida albicans*	1 (7.7%)	11 (84.6%)
*Mucor* sp.	1 (7.7%)	-
*Chaetomium* sp.	1 (7.7%)	10 (76.9%)
Total	11 (84.6%)	13 (100%)

**Table 5 diagnostics-12-02735-t005:** The pathogenic fungi identified by nanopore sequencing.

Patient	Pathogenic Fungi	Read Counts	Relative Abundance of Fungi (%)
1	*Malassezia* sp.	29	27.62%
	*Aspergillus* sp.	4	3.81%
	*Chaetomium* sp.	1	0.95%
	*Penicillium* sp.	1	0.95%
2	*Aspergillus* sp.	8	2.90%
	*Chaetomium* sp.	2	0.722%
	*Malassezia* sp.	105	38.05%
	*Penicillium* sp.	4	1.45%
3	*Aspergillus* sp.	1	0.8%
	*Chaetomium* sp.	1	0.8%
	*Malassezia* sp.	50	40.0%
4	*Aspergillus* sp.	9	2.91%
	*Candida albicans*	2	0.65%
	*Malassezia* sp.	100	32.36%
	*Penicillium* sp.	1	0.32%
5	*Aspergillus* sp.	56	2.52%
	*Candida albicans*	1	0.05%
	*Chaetomium* sp.	1	0.05%
	*Malassezia* sp.	309	13.93%
	*Penicillium* sp.	73	3.29%
6	*Aspergillus* sp.	20	2.96%
	*Candida albicans*	3	0.44%
	*Chaetomium* sp.	3	0.44%
	*Malassezia* sp.	304	45.04%
	*Penicillium* sp.	6	0.89%
7	*Aspergillus* sp.	11	3.17%
	*Chaetomium* sp.	1	0.29%
	*Candida albicans*	1	0.29%
	*Malassezia* sp.	137	39.48%
8	*Aspergillus* sp.	13	1.43%
	*Candida albicans*	4	0.44%
	*Chaetomium* sp.	3	0.33%
	*Malassezia* sp.	475	52.08%
	*Penicillium* sp.	11	1.21%
9	*Aspergillus* sp.	31	2.13%
	*Candida albicans*	6	0.41%
	*Chaetomium* sp.	5	0.34%
	*Malassezia* sp.	678	46.5%
	*Penicillium* sp.	11	0.75%
10	*Malassezia* sp.	112	17.67%
	*Aspergillus* sp.	16	2.52%
	*Candida albicans*	1	0.16%
	*Chaetomium* sp.	2	0.32%
	*Penicillium* sp.	17	2.68%
11	*Malassezia* sp.	94	34.43%
	*Aspergillus* sp.	3	1.10%
	*Penicillium* sp.	1	0.37%
12	*Malassezia* sp.	28	51.85%
13	*Aspergillus* sp.	20	4.26%
	*Candida albicans*	7	1.49%
	*Chaetomium* sp.	2	0.43%
	*Malassezia* sp.	58	12.34%
	*Penicillium* sp.	5	1.06%

## Data Availability

Not applicable.

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
