# Peer review of "Nasal Mycology of Chronic Rhinosinusitis Revealed by Nanopore Sequencing"

_diagnostics, 2022, doi:10.3390/diagnostics12112735_

Round 1

Reviewer 1 Report

This study emphasized the advantages of Nanopore sequencing in detecting fungi in CRS,

compared with traditional fungal culture method.

Comments:

1.     Why not set a normal control?

2.     This study showed traditional fungal culture successfully grew fungi in specimens of 11 (84.6%) patients, while the positive rate of fungal culture is much higher than reported in lots of articles, so I really want to know is there any special way or is there anything special about specimen selection?

3.     Patients with CRS were not categorized. Patients withs Polyps and non-polyps mixed with eosinophilic and non-eosinophilic types, which greatly increased heterogeneity. Moreover, three patients used the nasal steroids before enrollment, which may influenced the results.

4.     The size of Table 3 is not appropriate so that part of the text is hidden.

Reviewer 2 Report

Next generation sequencing techniques are a reproducible, affordable emerging diagnostic technique for mycosis of the respiratory tract (Brackin AP, Hemmings SJ, Fisher MC, Rhodes J. Fungal Genomics in Respiratory Medicine: What, How and When? Mycopathologia. 2021 Oct;186(5):589-608. doi: 10.1007/s11046-021-00573-x. Epub 2021 Sep 7. PMID: 34490551; PMCID: PMC8421194)

This article enhances the sensibility of this technique in detecting fungal genome in specimens from patients suffering from CRS: this topic is new in literature and its potential use in medicine could be very interesting. Maybe, the results of this study would have a higher soundness with a higher number of patients.

According to me, Authors should analyze specimens from patients with CRSwNP and CRS without NP separated.

Round 2

Reviewer 1 Report

“Using Ponikau et al’s method, Ponikau et al. (1999) cultured the nasal irrigant of 210 CRS patients and found a positive fungal culture rate of 96%, and Braun et al. (2003) got a positive fungal culture rate of 91.3% in 92 CRS patients.”

Comment:The method of fungal culture used in this article was proposed more than 20 years ago, and the other evidence given by the authors is from an article more than 10 years ago.   Aren't there any other similar articles in these years or is the method not clinically popular because it is time consuming and laborious?  In addition, what fungal culture method is routinely used at the authors' institution?
